

# Differing physiological and behavioral responses to anthropogenic factors between resident and non-resident African elephants at Mpala Ranch, Laikipia County, Kenya

Sandy Oduor[1], Janine Brown[2], Geoffrey M. Macharia[3],
Nicole Boisseau[4], Suzan Murray[5] and Paul Obade[3]

[1] Research, Mpala Research Centre, Nanyuki, Laikipia, Kenya
[2] Center for Species Survival, Smithsonian Conservation Biology Institute, Front Royal, VA, USA
[3] Department of Environmental Science, Kenyatta University, Nairobi, Nairobi, Kenya
[4] Endocrine Lab, Smithsonian Conservation Biology Institute, Front Royal, VA, USA
[5] Global Health Program, Smithsonian Conservation Biology Institute, Washington, DC, USA

Corresponding author
Sandy Oduor,
oduorsandy@yahoo.com

## ABSTRACT

**Background:** Heterogeneous landscapes like those of Laikipia County, Kenya consist of a mosaic of land-use types, which may exert differential physiological effects on elephants that occupy and traverse them. Understanding behavioral and physiological states of wild African elephants in response to the challenges of living in human-dominated landscapes is therefore important for conservation managers to evaluate risks imposed by elephants to humans and vice versa. Several conservation physiology tools have been developed to assess how animals respond to both natural and anthropogenic changes, and determine biological impacts. This study investigated how migratory and avoidance behavioral to vehicle presence, and vegetation quality affected fecal glucocorticoid (GC) metabolite (FGM) concentrations in African elephants at Mpala Ranch, Laikipia County, Kenya.
**Methods:** The study compared adrenal glucocorticoid activity of resident elephants that live within Mpala ($n = 57$) and non-resident elephants whose space use patterns overlap several ranches ($n = 99$) in Laikipia County, Kenya. Fecal samples were collected for a 4-month period between April and August for analysis of FGM concentrations. Behavioral reactions to research vehicles and body condition also were assessed. Satellite images from Terra Moderate Resolution Imaging (MODIS MOD13Q1) were downloaded and processed using Google Earth Engine to calculate a Normalized Difference Vegetation Index (NDVI) as a measure of vegetation quality.
**Results:** As expected, there was a positive correlation between avoidance behavior to vehicle presence and FGM concentrations in both resident and non-resident elephants, whereas there was an inverse relationship between FGM concentrations and NDVI values. Our study also found a positive influence of age on the FGM concentrations, but there were no relationships between FGM and sex, social group type, herd size, and body condition. However, contrary to our expectations, resident elephants had higher FGM concentrations than non-residents.

**Discussion:** Findings reveal elephants with stronger avoidance responses to research vehicles and resident elephants with relatively smaller home ranges exhibited higher FGM concentrations within the Mpala Ranch, Kenya and surrounding areas. Higher vegetative quality within the ranges occupied by non-resident elephants in Laikipia may be one reason for lower FGM, and an indication that the non-residents are tracking better forage quality to improve energy balance and reduce overall GC output. Additionally, our study found a positive influence of age, but no other demographic variables on FGM concentrations. Finally, adrenal glucocorticoid activity was inversely related to vegetative quality. Our findings can help conservation managers better understand how behavior and environment influences the physiological states of African elephants, and how management intervention might mitigate negative human–elephant interactions.

# INTRODUCTION

Over the past four decades, there has been a continuous decline in wildlife abundance in and outside of protected areas in sub-Saharan Africa (*Craigie et al., 2010*), including African elephants (*Loxodonta africana*) (*Wittemyer et al., 2014*; *Chase et al., 2016*). As a result, most large herbivore species continue to experience range reductions and local extirpation (*Ripple et al., 2015*). Many elephant populations also face threats from anthropogenic disturbances, including poaching, human–elephant conflict, land use changes, and habitat destruction (*Wittemyer et al., 2014*; *Chase et al., 2016*; *Thouless et al., 2016*), which can cause animals to alter movement patterns and/or flee areas in response to human presence (*Fahrig, 2007*). Behavioral indicators have identified effects of human disturbance in elephants (*Kiffner et al., 2014*; *McComb et al., 2014*), which can result in long-lasting behavioral changes, including altered responses to vehicles (*Vidya & Thuppil, 2010*). In other studies, movement patterns of wildlife in heterogeneous landscapes have been linked to reproductive success and fitness across spatiotemporal gradients (*Dingle & Drake, 2007*; *Owen-Smith, Fryxell & Merrill, 2010*; *Jachowski & Singh, 2015*; *Rolandsen et al., 2017*). In addition, studies of bioenergetics confirm animals that must travel greater distances to find adequate forage have lower fitness owing to decreased net energy intake, and reduced investment in reproduction and tissue maintenance (*Pontzer & Kamilar, 2009*).

African elephant movement patterns vary greatly, with some populations migrating over long distances, while others are more facultative migrants and become area residents (*Thouless, 1996*; *Dingle & Drake, 2007*; *Purdon et al., 2018*). The persistence of elephants in human-dominated landscapes raises important questions about the underlying factors motivating spatial behavior, data that could be used to inform on conservation and management planning to mitigate negative human–animal interactions (*Lima & Zollner, 1996*; *Nathan et al., 2008*). Foraging resource availability, predation risk, and competition (*Fryxell & Sinclair, 1988*; *Berger, 2004*; *Polansky, Douglas-Hamilton & Wittemyer, 2013*;

*Hopcraft et al., 2014*), and more recently, factors related to health and welfare are increasingly viewed as important to shaping animal distribution and movements (*Kyale, Ngene & Maingi, 2011*; *Goldenberg, Douglas-Hamilton & Wittemyer, 2018*; *Keigwin et al., 2018*; *Ihwagi et al., 2019*).

One factor that may be related to changes in spatial distribution of elephants is stress, which occurs when extreme and or prolonged physiological or behavioral adjustments are made in response to adverse environments to maintain homeostasis (*Dantzer et al., 2014*). The stress response typically involves activation of the hypothalamic-pituitary-adrenal (HPA) axis resulting in the release of glucocorticoids (GCs) from the adrenal cortex (*Bale & Vale, 2004*), which helps an animal cope with challenges and is generally adaptive (*Wingfield et al., 1998*; *MacDougall-Shackleton et al., 2019*). However, it is well documented that prolonged exposure to stressful conditions and elevated GCs can have negative, non-adaptive consequences, such as inhibition of reproduction, decreased growth rates, and immune suppression leading to increased disease susceptibility and decreased wound healing (*Sapolsky, Romero & Munck, 2000*; *Romero & Wingfield, 2015*). That said, it is important to note that GCs play an essential role in general energy metabolism, and are involved in normal physiological function as well (*Strack et al., 1995*; *Busch & Hayward, 2009*). In elephants, GCs increase during the follicular phase of the estrous cycle, pregnancy and parturition (*Brown & Lehnhardt, 1995*; *Fanson, Keeley & Fanson, 2014*). Thus, it is important to take biological context into account when relying on GC measures to evaluate stress and welfare.

The use of fecal samples to assess physiological responses to stressors, both positive and negative, via fecal glucocorticoid metabolite (FGM) analyses has become an important conservation tool to understand how animals respond to internal states and external environmental conditions (*Bradshaw, 2007*; *Denver et al., 2009*; *Soares et al., 2010*; *Ganswindt et al., 2012*). FGM monitoring has several advantages in that samples can be collected with minimal disturbance to the animal (*Harper & Austad, 2001*) and concentrations reflect pooled values over hours rather than a specific point in time, making them a better measure of overall adrenal activity (*Sheriff et al., 2011*).

The physiological demands of ranging over long distances can expose animals to stressful environments, making them more susceptible to disturbance (*Wilcove & Wikelski, 2008*). Negative influences of poaching, human-wildlife conflict, habitat loss, and population fragmentation can affect the physiology of elephants, including reduced reproductive output and increased FGM concentrations (*Foley, Papageorge & Wasser, 2001*; *Gobush, Mutayoba & Wasser, 2008*; *Graham et al., 2009*; *Hunninck et al., 2017*; *Vijayakrishnan et al., 2018*). FGM concentrations also have been linked to space use patterns in elephants; that is, the spatial refuge hypothesis, which predicts higher stress in animals with more restricted space compared to those with wider land use patterns (*Viljoen et al., 2008*; *Jachowski, Slotow & Millspaugh, 2012*; *Jachowski et al., 2013*). Finally, studies have shown that some wild African elephant populations exhibit higher FGM concentrations outside of protected areas (*Tingvold et al., 2013*; *Hunninck et al., 2017*; *Vijayakrishnan et al., 2018*), although not all (*Ahlering et al., 2013*).

This study examined FGM concentrations of elephants that utilize the Mpala Ranch, a 200 km$^2$ privately owned cattle ranch in the semi-arid savanna of Laikipia County, Kenya. Wildlife conservation on private ranches can play an integral role in ensuring the persistence of large mammals on land occupied by humans and livestock (*Hunt, 1997*; *Figgis, 2004*; *Cousins, Sadler & Evans, 2008*; *Sundaresan & Riginos, 2010*). This is in contrast to communal ranches where livestock concentrations are high, lethal methods are used to control wildlife, and human populations are increasing (*Kinnaird & O'brien, 2012*). Laikipia County is a non-protected savannah region (9,666 km$^2$) located on the equator that is divided into a mosaic of privately, publicly and communally owned ranches (*Graham, 2012*). Elephants within the Laikipia-Samburu ecosystem inhabit a wide variety of land uses with varying degrees of human occupation and disturbance (*Graham et al., 2009*; *Kahindi et al., 2010*), and little is known about how that affects ranging patterns and adrenal activity. Although the population has been increasing over the years (*Omondi, Bitok & Mayienda, 2002*; *Litoroh et al., 2010*; *Ngene et al., 2013*), Laikipia County has experienced land cover changes over the past three decades due to climate change and anthropogenic activities (*M'mboroki, Wandiga & Oriaso, 2018*), which may influence forage quality and elephant physiology. Thus, this study also examined how normalized difference vegetation index (NDVI)—a measure of forage availability based on the amount of red and near-infrared light from the earth surface (*Pettorelli et al., 2005*) potentially influences FGM concentrations in resident and non-resident African elephants at Mpala. We hypothesized that: (1) resident elephants with relatively smaller ranges on the private Mpala Ranch will have lower FGM concentrations than non-resident elephants due to high levels of human disturbance outside of Mpala; (2) elephants exhibiting avoidance behavior to research vehicles will excrete more FGM due to differences in coping strategies to environmental stressors; and (3) vegetative quality, measured as NDVI, will be inversely related to FGM concentrations due to higher nutritional stress during the dry season. This study also explored potential correlations between FGM and body condition, and a number of social and demographic variables, including sex, social group type (mother with calf, bulls, mixed male/female group), age group (juvenile, sub-adult, adults), and number of individuals in the herd, all of which have been shown to correlate with HPA activity in elephants and other mammals (*Foley, Papageorge & Wasser, 2001*; *Huber, Palme & Arnold, 2003*; *Rasmussen et al., 2008*). Understanding the links between ranging behavior, environmental quality, and the physiological state of elephants is important to monitoring elephant aggression towards people and increased levels of conflict in human-occupied landscapes for management interventions and species conservation.

## MATERIALS AND METHODS

### Study area

A map of the Mpala Ranch is shown in Fig. 1. It is a private landholding of 200 km$^2$ that supports wildlife conservation with limited livestock production. The property is not fenced and is surrounded by other private and communal ranches with differing degrees of wildlife tolerance. Mpala Ranch lies north of the equator on the Laikipia plateau (latitude,

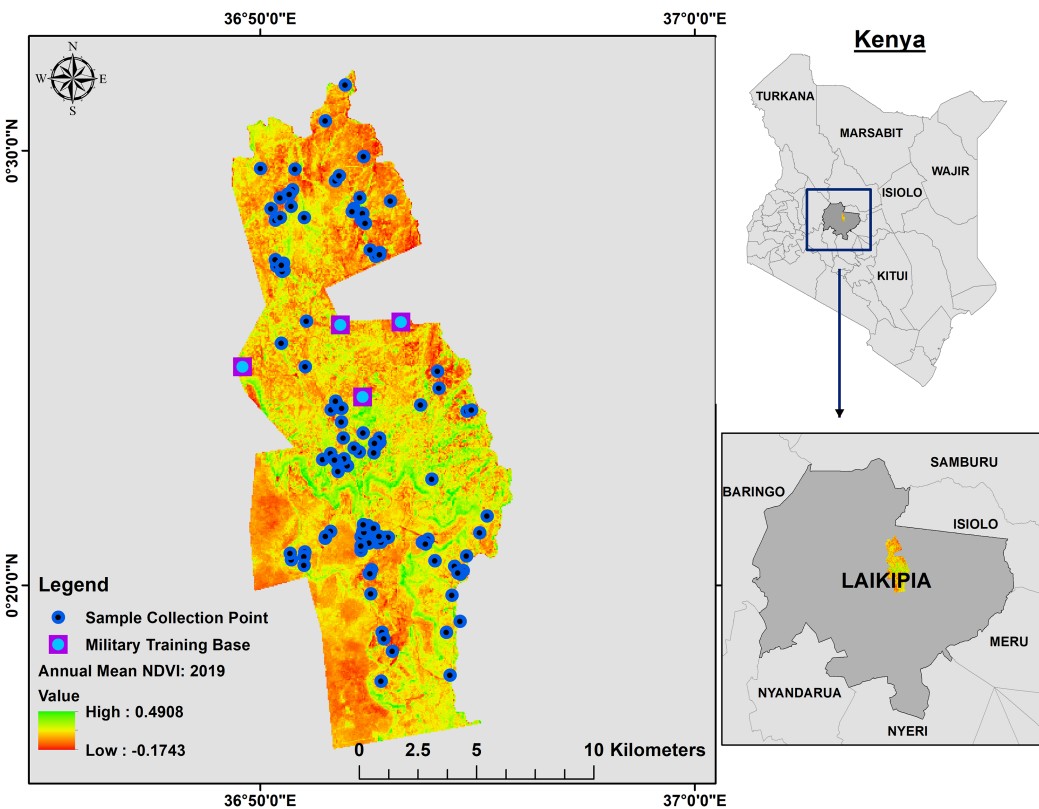

**Figure 1** **Annual mean 2019 NDVI base layer map of Mpala Ranch, Laikipia, Kenya showing where fecal samples were collected.** The map shows the distribution of fecal samples collection during the study period and where military trainings are conducted.

0° 17′ 32″N; longitude, 36° 53′ 52″E). Rainfall at Mpala Ranch averages ~600 mm per year in a weakly tri-modal annual pattern with a short dry season from December to March and peak rainfall in April–May and August–October (*Augustine & McNaughton, 2004*; *Pringle, 2008*; *Goheen et al., 2013*). The study area is characterized by woody vegetation, with *Acacia drepanolobium* being monodominant in black cotton vertisol soil (*Young et al., 1997*) and *Acacia brevispica, Acacia mellifera*, and *Acacia etbaica* species dominating the alfisol soil (*Pringle et al., 2016*). Other woody species include *Croton dichogamous, Rhus vulgaris* and *Grewia* species (*Young, Patridge & Macrae, 1995*). Although African elephants are general feeders, dietary preferences are dependent on season (*Codron et al., 2006*; *Kartzinel et al., 2019*), and movement behavior at Mpala Ranch is mainly influenced by woody cover, seasonal forage and water availability, and presence of human settlement and cattle bomas (*Ochieng, 2015*). In addition to African elephants, Mpala Ranch also is home to threatened lions (*Panthera leo*), leopards (*Panthera pardus*), cheetahs (*Acinonyx jubatus*), and endangered African wild dogs (*Lycaon pictus*). The area supports a number of large herbivores, including the endangered Grevy's zebras (*Equus grevyi*) and threatened reticulated giraffe *(Giraffa camelopardalis)*.

Between 2009 and 2014, an elephant monitoring program was conducted to understand how elephants use Mpala Ranch, assess their demographic characteristics (*Ochieng, 2015*),

**Table 1  Summary of study elephants at Mpala Ranch.**

| Age group | Ranging behavior | | | | Total |
|---|---|---|---|---|---|
| | Resident | | Non-resident | | |
| | Male | Female | Male | Female | |
| Juvenile | 14 | 7 | 19 | 3 | 43 |
| Sub-Adult | 9 | 6 | 10 | 6 | 31 |
| Adult | 0 | 21 | 23 | 38 | 82 |
| Total | 23 | 34 | 52 | 47 | 156 |

Note:
Demographic summary of elephants in the study monitored through behavior and fecal glucocorticoid metabolite (FGM) analyses from April to August 2019 at Mpala Ranch, Laikipia County, Kenya.

and determine proportions of resident versus non-resident families. Elephants are individually identified by ear and tusk features (*Douglas-Hamilton, 1972*; *Moss, 1996*) and classified as residents or non-residents based on the temporal use patterns (i.e., shift in abundance and degree of residency) throughout the ranch. Non-resident elephants are those that are observed intermittently at Mpala Ranch. Their movement in and out of Mpala Ranch is unpredictable, although in general they stay in the area between 3 and 5 months a year depending on vegetation quality and water levels around the dams at Mpala. Elephant home ranges typically overlap multiple communal and private conservancies within Laikipia (*Thouless, 1993*), whereas resident elephants are restricted to Mpala Ranch.

## Data collection

This study was conducted upon request and in collaboration with Kenya Wildlife Service (KWS) and National Commission for Science and Technology (NACOSTI), who issued the necessary research permits. A demographic summary of the study elephants at Mpala Ranch is shown in Table 1. Fecal samples and behavioral data were conducted between 23 April (toward the end of the dry season) and 30 August 2019 (toward the beginning of the short wet season). A pictorial database of the catalog recognition file developed by the Mpala Elephant Research Project was uploaded onto a Samsung Galaxy Tablet (T580 10.1) and used in the field for individual elephant identification. In the event that a new family group, lone bull or bachelor group were encountered, photos were taken using the method described by *Moss (1996)* and incorporated into the database. Examples of elephants in the catalog recognition file are shown in Figs. S1 and S2.

Elephants were located by driving on existing roads and off-road where possible, as well as through communications with security outposts throughout Mpala. When elephants were encountered, the behavioral reaction to the observer vehicle (i.e., reaction index) was categorized as follows: (1) nonresponsive within 20 m of vehicle approach (i.e., the activity of the family group or individual was not interrupted by approach of the vehicle within 20 m); (2) retreating by walking more than 20 m away from vehicle approach (i.e., the activity of the family group or individual was slightly interrupted by approach of the vehicle within 20 m but the elephants remained calm for at least 10 seconds); or
(3) retreating by running more than 20 m away from vehicle approach (i.e., the activity of the family group or individual was interrupted by the vehicle, and individuals responded by whipping the trunk or flapping its ears before running while vocalizing with tail raised). The vehicle's engine was then turned off and individuals were observed from a distance until defecation occurred. In the event an individual or family group ran more than 100 m from the vehicle or into a dense Acacia thicket and were out of visual range, no fecal samples were collected. Time of defecation, time of sample collection, sex, age group (juvenile = 0–8 years; sub-adult = 9–17 years; adult >18 years), social group (mother with calf, solitary bull, bachelor group, mixed male/female group), ranging behavior (resident or migrant), presence of temporal gland secretion in bulls, and number of individuals in the herd were recorded based on *Moss (1996)*. A body condition score (BCS) also was assigned using a 5-point scale developed for African elephants (*Morfeld et al., 2014*) with one representing very thin and five representing very fat. The coordinates where dung samples were collected were recorded using a Garmin 63 hand-held Global Positioning System.

## Assessment of vegetation quality

Vegetation quality within Mpala Ranch was assessed using a Terra Moderate Resolution Imaging Spectroradiometer (MODIS, MOD13Q1) that generated images every 16 days at a 250 m spatial resolution. Data were used to calculate NDVI (*Carroll et al., 2004*), a ratio of the near-infrared and red reflectance spectral bands that are reflected and captured by a satellite's sensor and scaled between 1 (which indicates green vegetation) and −1 (which indicates no vegetation) (*Pettorelli et al., 2005*). NDVI layers from MOD13Q1 were generated for the period between 23 April 2019 and 30 August 2019 to reflect the vegetation productivity and potential nutritional quality during the study. To account for phenological variation within the collected reference data, each fecal sample was linked to the NDVI data point for every location that had the closest available date. First, all the 156 coordinates represented by 156 fecal points were loaded onto the code editor (java) interface of Google Earth Engine (GEE). Satellite images were extracted online (https://lpdaac.usgs.gov/products/mod13q1v006) and defined to within Mpala where the fecal samples were spatially distributed. The satellite images were then filtered by masking pixels with cloud cover greater than 10% using the GEE simple cloud score function. Clouds, aerosol loading, shadows and other outliers, are known to affect the NDVI values and as a result removed using Savitzky-Golay smoothing filter (*Beltran-Abaunza, 2009*). NDVI was then calculated using the following equation: (near-infrared band − red band)/(near-infrared + red band). An NDVI chart panel for all the 156 points were then created based on the date of collection. Additionally, satellite images from Landsat 8 were mosaicked and clipped to within Laikipia County and also clipped to within Mpala Ranch. The annual 2019 NDVI base layer map for Laikipia and Mpala was compared as shown in Fig. S3.

## Fecal sample collection, processing and FGM analysis

One fecal sample was collected from 156 uniquely identified elephants within 2–30 min of defecation and stored in an insulated box with ice packs in the field. To account for

variation in concentrations of FGM among and within dung samples, at least two boli were broken open and subsamples taken from the interior part (Ganswindt et al., 2005). Samples were placed into 60-ml plastic tubes and labelled with a unique number, the date and family ID. Time from defecation to when samples were placed in a cool box in the field was recorded for each sample. Samples were then transferred to a −80 °C freezer at the Mpala Ranch within 6 h of defecation.

Fecal samples were extracted based on Wasser et al. (2000) at the Mpala Research Center Endocrinology Laboratory. Frozen samples were thawed, mixed thoroughly, and 0.50 ± 0.02 g extracted with 90% methanol in a 16 × 100 mm glass tube by vortexing for 30 min and then centrifuging at 1,800$g$ for 20 min. The supernatants were decanted into another set of 16 × 100 mm tubes and dried under air in a warm water bath. Dried extracts were reconstituted with 1 ml of assay buffer (Cat. No. X065; Arbor Assays, Ann Arbor, MI USA) and sonicated until completely re-suspended, then frozen at −20 °C until analysis.

Extracts were diluted 1:4 in assay buffer and analyzed in duplicate for FGM concentrations using a double-antibody corticosterone enzyme immunoassay (EIA) (Cat. No. K014-H5; Arbor Assays, Ann Arbor, MI USA) with no modifications. Optical density was determined using a plate reader (Multiskan FC; Thermo Scientific, Waltham, MA, USA) at 450 nm, and FGM concentrations were calculated using the sigmoidal dose response program (www.myassay.com). The EIA was validated for elephants by demonstrating parallelism between serial extract dilutions and the standard curve ($y = 1.03× + 5.13$, $r = 0.97$), and excellent recovery of unlabeled corticosterone standard added to a low concentration sample ($y = 0.95× + 49.25$, $r = 0.99$). A graph of parallelism and recovery curves are shown in Figs. S4 and S5 respectively. Biological validity was demonstrated by showing significant increases in FGM in an elephant 2 days before euthanasia for declining health (9.33 ng/g, $n = 1$) compared to pre-illness baseline (2.32 ± 0.14 ng/g, $n = 10$), before (1.19 ± 0.10 ng/g, $n = 12$) and in the days after (3.72 ± 0.14 ng/g, $n = 2$) a tail injury, and before (1.28 ± 0.16 ng/g, $n = 6$) and in the first week of a bout of lameness (3.72 ± 0.13 ng/g, $n = 3$). FGM concentrations using the Arbor Assays EIA also were correlated with samples analyzed by a previously validated corticosterone EIA (Watson et al., 2013) ($r = 0.94$, $n = 40$) and corticosterone radioimmunoassay (Wasser et al., 2000) ($r = 0.92$, $n = 30$). Based on our laboratory criteria, any duplicate with a coefficient of variation (CV) >10% or plate where the control CVs exceeded 15% of the control monitor were reanalyzed. In this study, no plates were redone and duplicates reruns were <5%. The EIA sensitivity was 0.90 pg/well (at 90% binding), and intra- and inter-assay coefficients of variation of high and low controls were 3.98% and 8.26% ($n = 7$ plates), respectively. Extraction and EIA analyses were carried out over a 3-week period starting 60 days after completion of fecal sample collection at the Mpala Research Center.

## Data analysis
FGM values were log-transformed to obtain a normal distribution of the model residuals. A linear regression model was used in constructing our model. Ranging behavior, reaction

index and NDVI were included in our basic model while the effect of each of the additional factors was included in the basic model to determine whether additional variables influenced FGM concentrations. Model selection was then done by comparing the basic model with the models consisting of both the basic model and additional factors that could be influencing FGM concentrations in African elephants. To discern differences in FGM concentrations among different factor variables *Emmeans* package (*Lenth et al., 2018*) was used. The competing models generated were ranked using ΔAICc <2. Akaike's Information Criterion adjusted for small sample sizes (*Burnham & Anderson, 2002*) to determine the model that was a better fit to the data in *MuMin* package. The basic model was then validated by plotting the results of the model to visually check for normality of the residuals, homogeneity, and the presence of outliers in the model (Fig. S6). Back-transformed model estimates were plotted to illustrate predicted response values and show relationships between the fitted response values and the predictor variables. Mean data are expressed as ± SD. Data were analyzed in R version 3.5.1 (*R Development Core Team, 2018*).

## RESULTS

Overall FGM concentrations ($n$ = 156) averaged 5.92 ± 2.15 ng/g with individual FGM ranging from 2.58 to 15.55 ng/g. FGM concentrations differed between sexes with females (6.30 ± 2.43 ng/g, $n$ = 81) having higher concentrations than males (5.51 ± 1.73 ng/g, $n$ = 75). Mean FGM concentrations were higher in adults (6.28 ± 2.28 ng/g, $n$ = 82) compared to juveniles (5.61 ± 1.63 ng/g, $n$ = 43) and sub-adults (5.39 ± 2.32 ng/g, $n$ = 31). FGM concentrations also varied among social group types with mothers with calf (6.22 ± 2.23 ng/g, $n$ = 103) having higher concentrations compared to bull groups (5.59 ± 2.37 ng/g, $n$ = 15) and mixed male/female groups (5.23 ± 1.68 ng/g, $n$ = 38). FGM concentrations varied by BCSs with BCS = 2 exhibiting higher FGM concentrations (6.90 ± 2.81 ng/g, $n$ = 18) compared to those with BCS = 3 (5.99 ± 2.14 ng/g, $n$ = 96) or BCS = 4 (5.33 ± 1.69 ng/g, $n$ = 42). There also was a difference in behavioral responses to vehicle presence between resident and non-resident elephants ($\chi^2$ = 22.13, df = 1, $p$ = $2.55^{-06}$, $n$ = 78) with non-residents reacting negatively to vehicle presences compared to the resident elephants. The month of June had the highest mean NDVI (0.522 ± 0.07), while April had the lowest (0.295 ± 0.11) (Fig. S7). Additionally, Laikipia County overall had relatively higher NDVI compared to Mpala Ranch (Fig. S5).

Our study found support for three models: the basic model that included factors related to ranging behavior, the behavioral reaction index and NDVI, and two models consisting of additional factors that might affect FGM, which were age (juvenile = 0–8 years; sub-adult = 9–17 years; adult >18 years) and social groups (mother with calf, solitary bull, bachelor group, mixed male/female group). There was, however, no influence of body condition, sex, herd size and time delay on FGM concentrations (Table 2).

In support of our hypothesis, ranging behavior, reaction index, and NDVI were all found to influence FGM concentrations ($F_{4, 151}$ = 9.4262, $p$ < 0.0001). Resident elephants exhibited significantly higher FGM concentrations (5.84 ± 0.5 ng/g, CI [5.27–6.48 ng/g]) compared to non-resident elephants (5.21 ± 0.6 ng/g, CI [4.60–5.89 ng/g]), while
**Table 2 Model selection based on the basic model and other factors influencing FGM concentrations in African elephants sampled at Mpala Ranch between April and August 2019.** Candidate models consisting of the basic model (in bold) and the models consisting of additional factors influencing FGM concentrations i.e. age group, group type, sex, herd size, BCS and time delay were ranked based on AICc. The models consisting of age group and group type were better fit than the basic model (in bold).

| Model | K | Model Likelihood | AICc | ΔAICc | Weight | $R^2$ | Adj. $R^2$ |
|---|---|---|---|---|---|---|---|
| FGM ~ Ranging behavior + Reaction Index + NDVI + Age group | 8 | −32.50 | 81.98 | 0.00 | 0.41 | 0.2347 | 0.2038 |
| FGM ~ Ranging behavior + Reaction Index + NDVI + Group type | 8 | −32.97 | 82.92 | 0.94 | 0.25 | 0.2300 | 0.2000 |
| **FGM ~ Ranging behavior + Reaction Index + NDVI** | **6** | **−35.97** | **84.51** | **2.53** | **0.12** | **0.1998** | **0.1786** |
| FGM ~ Ranging behavior + Reaction Index + NDVI + Sex | 7 | −35.02 | 84.79 | 2.81 | 0.10 | 0.2096 | 0.1832 |
| FGM ~ Ranging behavior + Reaction Index + NDVI + herd size | 7 | −35.80 | 86.36 | 4.38 | 0.05 | 0.2016 | 0.1750 |
| FGM ~ Ranging behavior + Reaction Index + NDVI + Time delay | 7 | −35.81 | 86.38 | 4.40 | 0.05 | 0.2015 | 0.1749 |
| FGM ~ Ranging behavior + Reaction Index + NDVI + BCS | 8 | −35.07 | 87.11 | 5.13 | 0.03 | 0.2091 | 0.1772 |

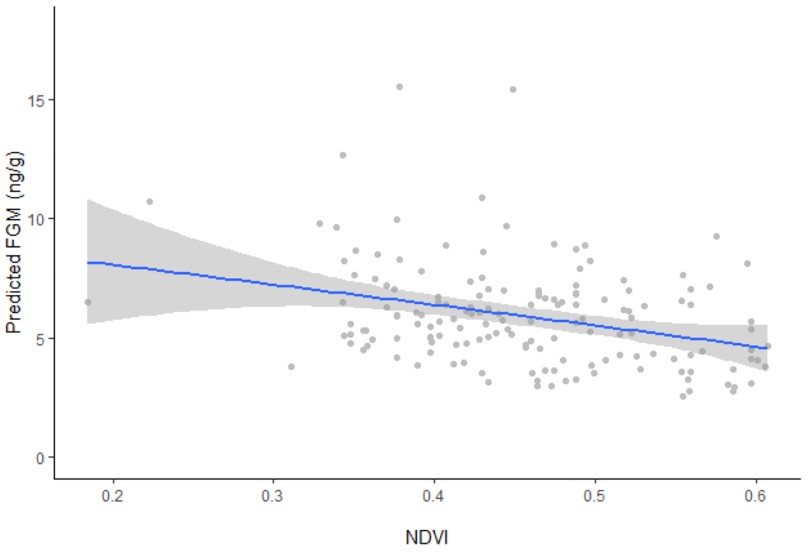

**Figure 2 Linear regression between FGM concentrations and NDVI with fitted data points.** Linear regression model plot with the 95% confidence interval (grey areas) showing the relationship between back transformed predicted fecal glucocorticoid metabolite (FGM) concentrations and the normalized difference vegetation index (NDVI) with fitted data points. Predicted FGM (ng/g) represents back transformed values of log FGM from the model.

concentrations were higher among individuals that retreated by running more than 20 m away from vehicle approach (6.07 ± 0.5 ng/g, CI [5.54–6.65 ng/g]) compared to those that were nonresponsive (5.21 ± 0.6 ng/g, CI [4.60–5.89 ng/g]) or retreated by walking more than 20 m away from vehicle approach (4.99 ± 0.4 ng/g, CI [4.61–5.40 ng/g]). Model results also showed an inverse relationship between predicted FGM concentrations and NDVI values (Fig. 2), with higher FGM concentrations observed when vegetation quality was low during the dry season (10.04 ± 1.6 ng/g, CI [7.38–13.64 ng/g]) compared to the wet season (4.23 ± 0.8 ng/g, CI [3.63–4.92 ng/g]) when Mpala had higher vegetation quality. Only age was found to be an additional factor influencing FGM

**Table 3 Results from model-averaged coefficients showing parameter estimates from the basic model and the two competing models.** The table consists of the basic model (Model 1), the model consisting of social group type (Model 2) and the model consisting of age group (Model 3) as additional factors influencing FGM concentrations in African elephants at Mpala Ranch.

| Coefficients | Model 1 | | | Model 2 | | | Model 3 | | |
|---|---|---|---|---|---|---|---|---|---|
| | Estimates | CI (95%) | P-Value | Estimates | CI (95%) | P-Value | Estimates | CI (95%) | P-Value |
| (Intercept) | 2.31 | [2.00–2.62] | <0.001 | 2.33 | [1.99–2.66] | <0.001 | 2.32 | [2.01–2.62] | <0.001 |
| Ranging behavior (Resident) | 0.12 | [0.00–0.23] | 0.04 | 0.09 | [−0.02 to 0.21] | 0.12 | 0.15 | [0.04–0.26] | 0.01 |
| Reaction Index (Retreating on vehicle approach) | −0.04 | [−0.17 to 0.09] | 0.52 | −0.04 | [−0.17 to 0.09] | 0.56 | −0.03 | [−0.16 to 0.10] | 0.60 |
| Reaction Index (Running away on vehicle approach) | 0.15 | [0.01–0.30] | 0.04 | 0.14 | [−0.00 to 0.29] | 0.05 | 0.15 | [0.01–0.29] | 0.04 |
| NDVI | −1.44 | [−2.07 to −0.82] | <0.001 | −1.5 | [−2.12 to −0.88] | <0.001 | −1.36 | [−1.98 to −0.74] | <0.001 |
| Social group type (Cow/calf) | | | | 0.06 | [−0.12 to 0.23] | 0.53 | | | |
| Social group type (Mixed) | | | | −0.09 | [−0.27 to 0.10] | 0.36 | | | |
| Age group (Juvenile) | | | | | | | −0.11 | [−0.23 to 0.01] | 0.06 |
| Age group (Sub-Adult) | | | | | | | −0.15 | [−0.28 to −0.02] | 0.02 |
| Observations | 156 | | | 156 | | | 156 | | |
| $R^2/R^2$ adjusted | 0.200/0.179 | | | 0.230/0.199 | | | 0.235/0.204 | | |

concentrations ($F_{6, 149} = 7.6143$, $p < 0.0001$). Although the addition of social group was a better model fit for the data ($F_{6, 149} = 7.4191$, $p < 0.0001$) than the basic model, it did not significantly affect FGM concentrations in the study population (Table 3).

# DISCUSSION

This is the first study to examine FGM concentrations in elephants living within the Laikipia county of Kenya and found relationships between FGM and migratory patterns, behavioral responses to vehicle approach, and vegetative quality. As predicted, elephants exhibiting avoidance responses to research vehicles had higher FGM concentrations. Similarly, higher FGM concentrations were found when vegetation quality, based on NDVI, was low. However, contrary to our expectations, resident elephants exhibited higher FGM concentrations compared to non-resident elephants, which may be related to higher vegetative quality in areas outside of Mpala Ranch before the onset of the wet season, or a variety of human activities in the region.

## Influence of ranging behavior on FGM

Based on other studies, we hypothesized that resident elephants with small home ranges would exhibit lower adrenal GC activity than long-distance migrants (*Wilcove & Wikelski, 2008*; *Pontzer & Kamilar, 2009*; *Wittemyer et al., 2017*). With the Laikipia-Samburu ecosystem experiencing high levels of poaching between 2007 and 2012, elephants have been observed to alter their circadian rest patterns in response to human activities (*Wittemyer et al., 2017*), increase their speed while traversing through risky landscapes (*Douglas-Hamilton, Krink & Vollrath, 2005*), and shift movement patterns to be more

active at night (*Ihwagi et al., 2018*). So we expected to observe higher FGM concentrations among the non-resident elephants. However, the opposite was true—resident elephants had higher FGM concentrations than non-residents. One reason could be exposure of resident elephants to higher levels of human activities in the northern part of Mpala. Earlier, *Awuor (2015)* observed a negative influence of military activities on the distribution and abundance of wildlife at Mpala. This training, primarily carried out in the north, involves drills and use of fire arms and explosives, which could have resulted in higher FGM concentrations in residents. In addition, the harassment of elephants by cattle herders around dams, particularly during the dry season when conflict over water resources is high, also could have affected adrenal steroid activity during the study period. Alternatively, elephants could have been responding to research vehicles and vehicular traffic along an arterial public road which cuts through Mpala Ranch, and if viewed as a threat could have resulted in higher FGM. Finally, our results lend support to the spatial refuge hypothesis, which predicts higher FGM concentrations in animals with more restricted space compared to those with wider land use patterns (*Viljoen et al., 2008*; *Jachowski, Slotow & Millspaugh, 2012*; *Jachowski et al., 2013*). Several studies have observed significant effects of human disturbances on FGM concentrations in elephants (*Gobush, Mutayoba & Wasser, 2008*; *Jachowski, Slotow & Millspaugh, 2012*; *Tingvold et al., 2013*; *Haase, Long & Gillooly, 2016*; *Hunninck et al., 2017*; *Vijayakrishnan et al., 2018*), and so all of these theories warrant further investigation.

*Jachowski et al. (2018)* observed the effects of high forage quality tracking and the use of stopover sites on the FGM concentrations of mule deer. Although elephant home ranges are restricted to both artificial and permanent water sources during the dry season (*Chamaillé-Jammes et al., 2013*; *Purdon & Van Aarde, 2017*), they are known to track peak forage quality (*Loarie, Van Aarde & Pimm, 2009*) to improve their energy balance which could have resulted in lower FGM concentrations in non-resident elephants at Mpala Ranch. Although the physiological demands of ranging over long distances exposes animals to stressful environments (*Wilcove & Wikelski, 2008*), access to better forage quality could act as a potential "pacifier" against stress (*Pokharel et al., 2019*) which then leads to lower FGM concentrations among non-resident elephants.

*Shaw & Couzin (2013)* posited that residency or non-residency behavior in animals is selected for when distribution of resources is dominated by local patchiness. Elephants are known to switch diets depending on seasonality (*Kartzinel et al., 2019*) and land use. Vegetative changes have occurred in Laikipia to a present day mosaic of grassland and Acacia bushland habitats (*Taylor et al., 2005*), and mixture of extensive agriculture, pastoralism and wildlife areas. Differences in FGM concentrations of elephants at Mpala Ranch could be attributed to non-residents having access to higher quality diets due to greater foraging, resulting in lower FGM concentrations. In India *Pokharel et al. (2019)* also observed lower FGM concentrations among crop raiding elephants, which was speculated to act as a mitigator against stress compared to elephants in protected forests with limited calorie-dense food sources.

## Relationships between behavioral responses and FGM

Results from our model revealed higher FGM concentrations in elephants that ran away from vehicle presence compared to those that were either nonresponsive or retreated slowly. However, although resident elephants exhibited higher FGM concentrations in general, the majority were either nonresponsive to or slowly retreated from vehicle presence compared to non-residents that generally retreated or ran. Studies have shown an influence of spatial knowledge and cognitive behavior on movement decisions of African elephants (*Polansky, Kilian & Wittemyer, 2015*). The difference in behavioral reactions could have been due to resident elephants being more habituated to research vehicles and not perceiving them as a threat and hence they did not respond as dramatically as non-residents. Older elephants retain memories of negative historical events (*Soltis et al., 2014*), which may disproportionately influence the behavior of the group (*McComb et al., 2001*). *Goldenberg et al. (2017)* observed differences in behavioral responses to vehicle presence between resident and non-resident elephants in Samburu and Buffalo Spring National Reserve in northern Kenya due to repeated exposure of resident elephants to researchers and tourist vehicles. Mpala Ranch also supports a number of activities with higher vehicular traffic from researchers and an arterial public road compared to neighboring conservancies, which could have influenced behavioral responses between resident and non-resident elephants at Mpala. It is important to note, however, that FGM is a measure of GC metabolite accumulation over 1–2 days prior to excretion, so we are not measuring real-time responses to vehicle presence, but rather the adrenal GC state of elephants that are more or less reactive using vehicle presence as a proxy.

The landscape of fear paradigm is an important mechanism related to the spatial ecology of animals across heterogeneous landscapes (*Gallagher et al., 2017*). The process by which an animal collects, stores and interprets environmental information not only influences their decision-making process, but also their behavior, physiology and life history (*Lima & Dill, 1990*). In other ungulates including elephants, studies have shown animals alter behavior by increasing flight distance in response to vehicles, human harassment or hunting (*Stankowich, 2008*; *Tarakini et al., 2014*; *Ranaweerage, Ranjeewa & Sugimoto, 2015*; *Szott, Pretorius & Koyama, 2019*), so the higher FGM concentrations could have been as a result of anthropogenic disturbance. Behavioral changes and increased physiological stress have also been observed in African elephants in response to disturbances caused by wildlife tourism (*Szott et al., 2020*). Finally, negative impacts on the behavioral response of wild Asian elephants to roads, vehicular traffic, and presence of heavy trucks have been observed in Asian elephants in Mudumalai Wildlife Sanctuary (*Gubbi, Poornesha & Madhusudan, 2012*; *Wadey et al., 2018*). Only small, four-wheel-drive vehicles were used in this study, which approached slowly to within 20 m of the closest elephant. Again, with a gut transit time of ~48 h (*Wasser et al., 2000*), immediate responses to vehicle presence would not be reflected in FGM concentrations. Even still, individuals exhibiting stronger avoidance behavior had higher FGM concentrations, an indication that negative behavioral reactivity to vehicle presence could be related to higher

adrenal activity overall. Negative effects of increased human population growth and pastoral settlements on wildlife distribution (*Ogutu et al., 2010*), and the recent subdivision and sedentarization of pastoral lands in savanna ecosystems that support livestock and wildlife (*Western, Groom & Worden, 2009*) shows there is need for both endocrine and behavioral monitoring to determine how human activities affect the stress physiology and fitness of wildlife.

## Influence of NDVI on FGM

Results from our model found an inverse relationship between FGM concentrations and NDVI at Mpala Ranch with both resident and non-resident elephants experiencing higher FGM concentrations when NDVI was low compared to when it was relatively high. Studies have used NDVI as a proxy for forage quality and dietary protein (*Hamel et al., 2009*; *Petorelli et al., 2011*; *Stabach et al., 2015*; *Hunninck et al., 2020*), and its use to assess forage quality in relation to FGM concentrations in wildlife is on the rise. For instance (*Stabach et al., 2015*) observed an inverse relationship between FGM concentrations and NDVI in wildebeest in Kenya. *Pokharel et al. (2019)* also confirmed an inverse proportionality between FGM concentrations and NDVI in free-ranging Asian elephants in India, while *Hunninck et al. (2020)* found an inverse relationship between FGM concentrations and NDVI in impalas in the Serengeti ecosystem. Our study was limited by having fewer samples collected during the dry season (end of April to mid-May); nevertheless, our findings confirm the influence that vegetation quality potentially has on the adrenal response of wildlife. The fact that migrant elephants had lower FGM concentrations than residents suggests they may be tracking vegetation quality to improve bioenergetics. A wide-ranging foraging strategy may better meet nutritional demands (*Parker, Barboza & Gillingham, 2009*; *Sach et al., 2019*), thus reducing GC output.

## Other factors influencing FGM

Our study found a relationship between FGM and age group, with adults exhibiting higher FGM concentrations than other age categories, similar to what was observed in free-ranging Asian elephants (*Vijayakrishnan et al., 2018*). This could be due to increased responsibilities and other leadership roles adults play within the herd (*McComb et al., 2011*). However, other studies have found no clear relationship between FGM concentrations and age in free ranging elephants in other parts of Africa (*Ganswindt et al., 2005*; *Viljoen et al., 2008*). From our model, the variable social group type did not influence FGMs, although mothers with calves did exhibit significantly higher FGM concentrations compared to mixed groups. The high FGM concentrations in mothers with calves could be attributed to their increased level of "awareness" in heterogeneous landscapes and areas that pose higher poaching risks or dangers to the calves (*Boettiger et al., 2011*), and their role in caring for other calves (allomothering) in the herd (*Lee, 1987*). Other authors have also found a positive relationship between lactating mothers and FGM concentrations in free-ranging Asian elephants (*Pokharel, Seshagiri & Sukumar,*

*2020*). Otherwise, there were no differences in FGM concentrations between mothers with calves and bulls or between bulls and mixed groups.

Sex differences in hypothalamo-pituitary-adrenal responses have been observed in multiple species, and may be related to social and physiological factors (*Creel et al., 2013*). In elephants, some studies have found higher FGM concentrations in females than males (*Hunninck et al., 2017*), while others show the reverse (*Ahlering et al., 2013*), and there are additional examples of no differences between the sexes (*Munshi-South et al., 2008*; *Tingvold et al., 2013*). Our data agree with the latter, with no significant difference in FGM concentrations between sexes. Thus, whether physiological or behavioral variation explains sex differences, or if it is due to other extrinsic factors, remains to be determined.

Although our model found no influence of BCS on adrenal activity in the elephants at Mpala Ranch, individuals with lower BCSs had relatively higher mean FGM concentrations compared to individuals with higher scores. Low BCS has been associated with nutrient-deficient diets and resource limitations, resulting in increased protein catabolism and muscle deterioration, and subsequently elevated stress levels (*Harvey et al., 1984*). In a study of wild Asian elephants, a similar relationship between high FGM and low BCS was found and thought to be due to seasonal shifts in diet quality (*Pokharel, Seshagiri & Sukumar, 2017*). Finally, *Mumby et al. (2015)* observed seasonal changes in body weight and FGM of logging elephants in Myanmar related to rainfall (body weight) and the work period (FGM). In our study, data were only collected for 4 months, so it is possible a longer collection period would reveal clearer relationships between body condition and adrenal function.

African elephants exhibit a fission-fusion social society (*Douglas-Hamilton, 1972*; *Moss & Poole, 1983*; *Wittemyer, Douglas-Hamilton & Getz, 2005*; *Couzin, 2006*), with benefits of congregating in large groups include cooperative defense of calves, and shared social and ecological knowledge from older, more experienced herd members (*Dublin, 1983*; *McComb et al., 2001*). However, our study found no relationships between social group size and FGM concentrations, or the overall response to vehicle presence. This study also did not find any relationship between herd size and FGM concentrations in relation to NDVI. Thus, while other studies have found higher FGM concentrations associated with larger herd sizes, particularly during the dry season (*Foley, Papageorge & Wasser, 2001*), our study was only conducted for 4 months, so additional annual analyses are needed.

The influence of time delay between defecation and sample stabilization via freezing on the degradation of hormone metabolite concentrations is an important consideration (*Möstl & Palme, 2002*). *Hunninck et al. (2017)* found a slight increase in FGM concentrations with increasing time delay, at about 2 h post defecation in African elephants. *Webber et al. (2018)* determined that hormone degradation starts to occur at about 20 h after defecation in African elephants. All samples in our study were collected within 30 min after defecation and placed in a cold box before being frozen within 6 hours, so it was not surprising there was no relationship between FGM concentrations and delay time.

## CONCLUSIONS

This study found that elephants with strong avoidance responses to research vehicles and resident elephants with relatively smaller ranges exhibited higher FGM concentrations within Mpala Ranch in Kenya. This information is important and can inform policy makers like the Kenya's National Elephant Management and Conservation Strategy and conservation managers on how elephant physiological state is influenced by ranging and foraging behavior in human-occupied landscapes. With it, we can begin to identify land use types that are compatible or incompatible with conservation, and establish wildlife corridors to avoid conflict. Due to unpredictable weather patterns, more sampling was carried out during the wet compared to the dry season when conflict with herders is usually more intense, so physiological reactions may have been under estimated. Thus, there is need for a more comprehensive, multi-year studies to determine how migratory behavior and vegetation changes affect the physiological stress response of both resident and non-resident elephants in the region.

## ACKNOWLEDGEMENTS

We sincerely acknowledge Fardosa Hassan for her support in acquiring research permits and procuring stationery required for the project, and Dr. Dino Martins, Director at Mpala Research Centre and his staff, and the ranch security guards for support of the study. We acknowledge Margaret Kinnaird, Practice Leader at WWF International for her reviews and comments on earlier versions of the manuscript. We thank Save the Elephant (STE), who funded the Mpala Elephant Monitoring Program between 2009 and 2014 and whose catalogs were used in this study to identify resident and non-resident elephant families.

### Funding

This work was supported by the National Geographic Society (grant number EC-55779R-19) and Idea Wild, a registered 501c (3) non-profit organization. The funders had no role in study design, data collection and analysis, decision to publish, or preparation of the manuscript.

### Grant Disclosures

The following grant information was disclosed by the authors:
National Geographic Society: EC-55779R-19.
Idea Wild: 501c (3).

### Competing Interests

The authors declare that they have no competing interests.

# PeerJ

## Author Contributions

- Sandy Oduor conceived and designed the experiments, performed the experiments, analyzed the data, prepared figures and/or tables, authored or reviewed drafts of the paper, and approved the final draft.
- Janine Brown conceived and designed the experiments, performed the experiments, analyzed the data, prepared figures and/or tables, authored or reviewed drafts of the paper, and approved the final draft.
- Geoffrey M. Macharia performed the experiments, authored or reviewed drafts of the paper, and approved the final draft.
- Nicole Boisseau performed the experiments, analyzed the data, authored or reviewed drafts of the paper, and approved the final draft.
- Suzan Murray performed the experiments, analyzed the data, authored or reviewed drafts of the paper, and approved the final draft.
- Paul Obade performed the experiments, analyzed the data, authored or reviewed drafts of the paper, and approved the final draft.

## Field Study Permissions

The following information was supplied relating to field study approvals (i.e., approving body and any reference numbers):

Field experiments were approved by Kenya Wildlife Service (KWS) (Ref. No. KWS/BRM/5001) and National Commission for Science and Technology (NACOSTI) (Ref. No. NACOSTI/P/19/45794/28283).

## Data Availability

Raw data and codes are available in the Supplemental Files.

## Supplemental Information

Supplemental information for this article can be found online at http://dx.doi.org/10.7717/peerj.10010#supplemental-information.

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
