# Peer review of "Differing physiological and behavioral responses to anthropogenic factors between resident and non-resident African elephants at Mpala Ranch, Laikipia County, Kenya"

_PeerJ, doi:10.7717/peerj.10010_

## Round 0.1 · original submission · Major Revisions

I have been extremely fortunate to receive very detailed and useful reviews from three reviewers that are far more expert on this topic than I am. Thus, I am going to defer to their comments which converge on the opinion that your study is generally well-done and will make a nice contribution to the literature. However, each of the reviewers has highlighted the need to be more careful and specific in your use of terminology; particularly in how you characterize glucocorticoids, human disturbance and migratory status. The reviewers also make some suggestions for improving the statistical analyses and for focusing on some data that you gloss over. Please address the reviewers' comments carefully in your revision making changes directly to the manuscript.

Reviewer 1 ·

Basic reporting

Professional English: Acceptable. Some minor grammatical and syntax errors have been noted in General Comments below.

References & background: Up-to-date references are cited as appropriate

Professional article structure w raw data: Clear and classic structure; raw data appear well organized and understandable

Self-contained unit of publication: Yes

Experimental design

Within aims & scope of journal: Yes

Research question well defined, relevant & meaningful, fills a gap: Yes

Rigorous, high technical standard, high ethical standard: Yes - this is quite a strong study.

Methods described in sufficient detail: Some additional details are necessary for assay methods, which I have noted in General Comments below.

Validity of the findings

Data provided; robust, statistically sound, controlled: Yes

Conclusions well stated, linked to original question, limited to results: Some terminology is fuzzy/subjective and needs clarifying, particularly the quating of FGMs to "stress" and the subjective terminology used for describing elephant reactions to vehicles.

Speculation is identified as such: Yes

Additional comments

This is a nicely done study that focuses on fecal glucocorticoids of African elephants, essentially using the fecal GC's as a proxy for physiological stress. Sample size is impressive and the authors have also managed to gather data on a solid battery of seven variables that may all affect physiological stress and hence fecal glucocorticoids: vegetation quality, body condition, migratory behavior, response to human vehicles ("terrified" or not), sex, demographic group, group size. The use of landscape-scale satellite-based NDVI data is especially welcome, allowed the researchers to discriminate forage quality effects from human disturbance effects. An additional strength is the inclusion of body condition scores. Finally, the use of behavioral reactions to a vehicle as a proxy for human disturbance is quite interesting way to assess potential exposure to past poaching. It is not entirely novel (flight distance has been studied in other species) but combining it with cort data is new, to me at least. It is a strong and useful study, and is quite well-written.





Some limitations, however, should be addressed or mentioned:
- The only major limitation of experimental design is that the study was limited to a single four-month period of a single year (April-August of one year), and thus it is not clear if the patterns reported here are generalizable to other seasons, other years, or of course to other locations. I recommend the authors make this limitation clearer in the Discussion and perhaps soften their main conclusions accordingly.

- Terminology needs to be made more rigorous and objective, particularly (1) the conflation of "glucocorticoids" with "stress" (see specific comments below) and (2) use of the highly subjective word "terrified" to describe elephants fleeing a vehicle. They may well be terrified, but we cannot ever truly know that. The standard in animal behavior is to describe a behavior by what it looks like, not by what we think it means or what we think the animal is feeling. I recommend all instances of "terrified" be deleted and replaced with a description of what the animal is actually doing (running away from vehicle). See specific comments for details.

- Most of the paper is well structured but I think the flow of the Introduction could be improved. The paragraphs in this section seemed to lack logical connectivity to each other. I recommend inspecting the transition points between paragraphs (compare last sentence of the prior paragraph and the first of the next, to see if the topics actually connect logically). One idea is to swap the Laikipia County study-area paragraph with the stress/reproduction paragraph. Also, some sentences could be deleted (example: lines 63-64 - we never hear again about calves, nor about energetic costs)

- The authors are understandably focused on the fleeing-from-vehicle behavior, but as a result it seems that they downplay a lot of their other results unnecessarily. The body condition data, for example, is fascinating but is treated a bit dismissively as just a "confounding" factor. The authors also have 3 cases of injuries/illness that are a lovely physiological validation. I think the scope of the whole paper could be slightly broadened to feature some of the other results a bit more strongly. This would involve only minor rewording.

- The hormone assay section needs some more methodological details, particularly on QA/QC criteria.

Additional specific comments are as follows

Line 30 - minor edit, I believe "human" should be "humans"?

Line 57 - missing word, "As result" should be "As a result"

Line 69 - consider "tissue maintenance" instead of "general maintenance" ("general" is vague)

Line 71 - "transitory" seems not the right word to contrast with long-distance migrants. Long-distance migrants are also transitory (they transit the landscape). Are you referring to short-distnace migrants, or nonmigratory individuals, or facultative migration?

Line 83 - "inversely proportional...across three main land use types" is a strange grammatical construction. Inversely proportional to what exactly? (If you mean human use, say so = "inversely proportional to human use across three main land use types")

Line 84-85 - it is not clear how land use #1 compares to land use #2. Do ranches have more cattle than the pro-wildlife properties? Are there cattle on #3 as well? It is not clear whether you are viewing these 3 land use types as 3 different intensities of human disturbance, 3 different densities of cattle maybe, or just 3 qualitatively different categories.

Line 88-89 - I tripped up on the "due to difficulty in..... and maintaining" syntax and had to re-read. Maybe trying adding another "difficulty in" (due to difficult in ensuring.... and difficulty in maintaining...")

Line 92 - not clear if the PIKE data is from Laikipia specifically or from a larger region; clarify

Line 94 - not clear if "the ecosystem" refers to Laikipia or a larger area; clarify

Line 102-121 - A major terminology issue with the entire paper is the conflation of glucocorticoids with "stress". Glucocorticoids are fundamentally about fuel availability, not stress per se. It's certainly common shorthand for wildlife biologists to refer to GCs as "stress hormones" but in a formal paper it's really better to just call them "glucocorticoids", be cautious and sparing about where you use the word "stress" and keep it very clear for the reader what you are actually measuring. For instance in line 116 ("some elephant populations exhibit more stress") it's not at all clear if by "more stress" you mean elevated GCs, elevated heart rate, disrupted behavior, poor reproductive performance or what exactly. If you meant GCs there, say GCs. See "Glucocorticoids and stress are not synonymous" (MacDougall-Shackleton et al., Integrative Organismal Biology 2019, doi :10.1093/iob/obz017).
Additionally there are the complications that (a) GCs elevate during "good stress" as well, two prime examples being migration and mating activity, neither of which are bad things; and (b) GCs can decline in cases of long-term chronic stress. As one final complication, glucocorticoids were not actually what was measured; this study measured fecal glucocorticoid metabolites, and there is ample literature showing that FGMs are a imperfect measure of plasma GCs. FGMs can be affected by fiber content, diet switching, gut microbiota, volume of food processed, individual differences in urinary vs. fecal excretion routes, dramatic differences in metabolite binding affinity to assay antibodies, etc. In the end I do agree it's reasonable to use FGMs as a proxy for GCs, and GCs in term as a proxy for physiological stress. In fact there is a strong literature justifying this approach. But we are using a proxy of a proxy in the end, the data are inevitably very noisy, and the authors have not made clear that they are fully cognizant of all of these issues. I suggest adding a short section to the Intro & a few sentences to the Discussion to alert the reader to these issues and better defend the approach. I also recommend searching the entire text for every instance of the word "stress," and changing as many as possible to clearer terminology ("GCs", "adrenal hormones" or the like).

Line 122 - It's overstating your data to say you "examined the physiological states". FGMs alone are not a physiological state. Consider just saying you examined fecal GC metabolites.

Line 124 - It is not clear to me why you hypothesized that residents would have lower FGMs. Explain the logic for hypothesis #1. (edit: later I found the rationale in the Discussion. Put it here in the Intro too.)
It would also be nice if hypotheses 2 and 3 were slightly fleshed out (adding a phrase) to delineate their logic as well - i.e. I think you are implying that #2 is really about past anthropogenic stress, while #3 is about current nutritional stress (I assume?)

Line 13 - "degree" should be "degrees"

Line 141 - "and" and "species" should not be italicized

Line 142 - "and" should not be italicized

Line 149 - "tusks" should be "tusk"

Line 156 - extra comma after "families" can be deleted

Line 157-159 - This sentence should be in the next section since it is not general background about the population; it is about the specific elephants from which you collected data. Also, I think there is a word missing from the start of the sentence? ("A demographic summary"?)

Line 161 - Explain what season April-August is - was this dry season or wet season, or does this area even have dry/wet seasons? (edit: found this later in the paper - mention it here too.)

Line 170 - The idea of using behavior response to vehicles as an indicator of prior disturbance is really interesting, but I think needs to be set up better in the Introduction. Perhaps add a sentence to one of the Intro paragraphs that already mention human disturbance, saying that elephants exposed to significant human disturbance (especially poaching, I assume?) are known to develop long-lasting behavioral changes that include response to vehicles.
Also - am I correct in assuming that vehicle avoidance probably indicates poaching specifically, not other types of anthropogenic disturbance? (i.e. I'm thinking that cattle ranching per se, or road traffic per se, would not induce vehicle avoidance behaviors?) Clarify that somewhere (probably in the Discussion)

Line 172 - Describe what you mean by "skittish." Describe what the animal actually does, not what you assume it is feeling; ears out? tail up? rapid locomotion? What exactly do you see?)
Line 172 - Delete "terrified" - we can't know what they are feeling. (They might be terrified, they might furious, they might be annoyed; personally I suspect they are indeed scared, but we can never know.) The golden rule when describing animal behavior is to name it by what it actually looks like, not what you think it means. In this case: say, rapid running away from the vehicle (or speed-walking or however is the best way to describe elephants moving fast - they don't really do a true run). Add any observable details that helped you identify this state: do they vocalize? Were you also looking at body language (ears out, tail up, trunk posture etc.)? Is fast locomotion alone enough to qualify as #3?
Additional question: Did any elephants flee entirely so that you lost sight of them and never got a fecal sample? I am wondering whether you may have missed sampling of the highest-GC elephants simply because they chose to remove themselves from your study! Could this have impacted your dataset?

Lines 174-175 - Are these ages known exactly from sightings records (were all the elephants first sighted as calves), or estimated from body size or other attributes?

Lines 177-179 - Which end of the BCS scale is skinny and which is fat? Readers need to know this to make sense of the BCS data later.

Lines 184-185 - both instances of "indicate" should be "indicates"

Line 185 - Mention in the paragraph above that you got GPS location of all the poops (this wasn't mentioned until now)

Line 188 - comma missing after "platform"

Line 190 - I don't understand this ("filtering the GPS locations based on the date of sample location") Do you mean something like, each fecal sample was linked to the NDVI data point for that location that had the closest available date? What did you do if a fecal sample was collected on a date that was right in between two NDVI dates?

Line 191 - Did any elephants contribute more than 1 fecal sample to the dataset? That is, did you have exactly 156 fecal samples (one per elephant)? - if not, explain any anomalies. Assure that reader that pseudoreplication was not an issue (unless it was! In which case, fix it - use only 1 sample per individual or average the results of multiple poops)

Line 199 - Where did extraction and assay take place? You have not described sample shipment so it comes across like the assays all occurred at Mpala, which I doubt is the case. If there was a sample shipment step, clarify how samples were shipped (dry ice? ice packs?) and where the labwork was done.

Line 200 - Clarify the precision of the scale used. Do you mean it was only precise to within 0.02g, or that you accepted weights within a range of 0.48-0.52 but weighed them to (say) the nearest 0.0001g (or whatever)?

Line 205 - State how many months elapsed in total between collection and final analysis (assay).

Line 207 - space missing between "enzyme" and "immunoassay" (it is two words, not one)

Line 207 - some necessary detail is missing about the assay and QA/QC protocols:
- Did you do a matrix effect test? (spiking a set of standards with fecal extract, comparing them to unspiked standards) It sounds like you did 1 such sample but did you do it with the entire standard curve? (you would then graph apparent dose vs. known standard dose and look at the slope - it should be near 1)
- Did you add any extra low-dose standards? (common for this assay)
- Was a standard curve run in every assay?
- Were controls in every assay? Were they species controls, pure hormone controls, or what?
- Were samples, standards and controls run in duplicate or triplicate?
- Were samples from your two groups (resident and transient) divided equally across and within assay plates, or randomized? That is, did you take steps to ensure that normal assay variation did not cause a spurious confounding effect in your data? (example: it would be bad to run all transient elephants in one plate and all resident elephants in another plate.) Generally - state the # of separate assays run (I see later that it may have been 7?) and how you arranged your samples across the assays. In this case of a single-year study, you might also be able to state whether all assays used the same lot #s of reagents, which is always nice.
- Important: What QA/QC criteria determined whether data were acceptable or whether a sample would need re-running? For example, common screens would include: any sample with high %CV is rerun (state the threshold), samples outside a predetermined acceptable range of %bounds would be rerun (state the allowed range), assays with anomalous controls or standards would be rerun, etc. Describe whatever methods were used to ensure quality control of assay data, and how re-run and re-dilution decisions were made.

Line 210-211 - If this assay has not been validated for elephant feces before, you need to provide all details - statistical results of the parallelism test, and I'd recommend putting the parallelism graph into supplementary data.

Line 211 - "significant" is probably not the best word here; try "acceptable" or even "excellent"

Line 212-215 - I would break out these validations and feature them more prominently; have the issue of validation be one of your study objectives, describe each case in more detail here, and move the FGM results to their own little paragraph in Results. It's quite nice that you found three cases of injury/illness; consider describing them a little more. (Is cause of death known for the elephant that died?)

Line 222 - commas missing around the dependent clause ("The response variable, i.e. FGM concentrations, was first...")

Line 231 - comma missing at the end of the dependent clause (after "(2018)").

Line 223 - Bates et al. should be inside the parentheses

Line 233-237 - There are two sentences here that need to be moved to Results - the one that states the best model, and the last sentence about "confounding" variables. (edit: Now that I have read Results, I think these may be redundant and could be deleted? At any rate, they should not be in Methods)
And btw I would also recommend deleting the word "confounding" - "confounding" does not just mean another significant factor. They're additional factors that significantly affect cort, but none of them are not confounding your ability to determine the effect of other variables.

Lines 250-258 - The Hypothesis Testing paragraph is in between two other paragraphs that each are descriptive data with means and SEMs. This doesn't really flow that well - I suggest putting the modeling results either at the beginning of Results or at the end.

Line 262 - Again, delete the word "terrified" here, and from the table about model results too, and from the figures.

Line 273 - suggest rephrasing to say: "relationships of FGM with"

Line 274 - delete "fearful", replace it with something descriptive that does not attempt to state what the elephant's (unknowable) emotions might be. "Avoidant" perhaps.

Line 276 - It is not clear why your expectations were that residents would have lower cort, particularly given that you know there is "considerable anthropogenic activity" in Mpala.

Lines 276 and 282 - the phrase "However, contrary to our expectations" occurs in both these lines. Eliminate the redundancy

Line 309 - reword to eliminate "terrified"

Line 335 - word missing, "there is a need"

Line 355 - as mentioned above, I think these are not really "confounding" factors. They are additional factors. (A true confounding factor is one that correlates unexpectedly with one of your key variables in such a way as to eliminate your ability to answer your main question. For example: if all residents were female and all transients were male, in that case sex would be a confounding factor because it would eliminate your ability to determine the true effect of residence/transience.) Just delete "confounding" and make a new heading, something like "demographic effects"

Line 357 - were these mothers still nursing? If so, lactation could explain the high cort.

Line 363 - replace "stress" with FGM

General comment on discussion: This is a nicely written Discussion that moves logically through all your findings, including effects of sex, groups size, BCS. However a lot of these topics were not mentioned in the Intro. Every topic in the Discussion should be mentioned, at least very briefly, in the Intro. You have a strong study here in that you took the trouble to quantify at least seven different factors that could affect fecal cort: available forage (NDVI), body condition, landscape use (residence/transience), prior human disturbance (reaction to vehicle), sex, group size, group type. It struck me as I read through the Discussion that you may have unnecessarily downplayed many of these variables. In essence this is a study that is stronger and also broader than it is described to be. Consider broadening out slightly. I suggest not referring to the other variables as "confounding", elevate them in your mind to major and interesting effects, and mention them in the Intro. This doesn't need to be a huge rewrite, just adding a few select sentences to the Intro to alert the reader that these other variables exist and will be explored. For example, in the last paragraph of the Introduction you could add something like "we also explored potential correlations of cort with body condition as well as with social and demographic variables XYZ, as these factors have been shown to correlate with HPA axis activity in numerous mammals" or whatever.

Thought on the flight behavior: Could elephants learn the running-away behavior from each other? For example, could a calf pick it up from its mother? Maybe add a comment on whether this might be a behavior that elephants learn from herdmates or immediate family. This is relevant for whether that specific elephant really encountered poachers, or whether it's more that the population in general experiences poaching.

Line 398 - replace "fearful" with a more objective term

Lines 405-407 - You need to acknowledge somewhere the substantial limitation to this study, which is not only that it was limited to one season, but also to one year. This would be a good place to put such a caveat (not just the limitation to one season, but also that it was only in one year). Then in the next sentence say "multi-year", not "year-long". All studies should include more than one year whenever possible since there are almost always powerful year-specific effects ("bad years" and "good years", etc.) that can distort or amplify hormone patterns. It's hard to know how generalizable your data are without additional years, so make sure you acknowledge this explicitly and make a concrete recommendation that future studies should span multiple years.

Figure 1 - I can't see the KLEE plots or the Uhuru block. Are they the black dots in lower left? You may need to make the black lines thinner.

Table 1 - It would be good to see the sex ratio within the non-resident group and within the resident group - I can't tell if sexes were unequally divided across the two groups. Perhaps divide those columns into the two sexes, or add the sex division in parentheses.

Figures S1 and S2 are great but they don't seem to have a figure legend? Perhaps mention the some of the features that enable identification (ear margin, etc.)

Thank you for doing this interesting and valuable study.

Reviewer 2 ·

Basic reporting

Manuscript is very well written, clear, and easy to read (unusually so). I have some minor suggestions for improvement and contextualization, and I think that Fig. 3 should show the data points underlying the regression.

Experimental design

Study design is strong. Replication is high. Results withstand common-sense scrutiny. I suggest one additional analysis, which I didn't see anywhere and think is important for the authors' interpretation of their results. Specifically, because the interpretation is that human disturbance especially in the north of the study area may help to explain the greater stress of resident animals, I think it would be useful to assess whether there is any trend in FGM from north to south within the study area. The data set appears to allow this, as samples were collected across the property. The analysis would have to control for potentially confounding factors, notably NDVI, which should be higher in the south.

Validity of the findings

Findings are valid, to the best of my ability to assess.

Additional comments

This is a timely, well written, and well executed study. Its results contribute both to our general understanding of how human activities influence the behavior and physiology of African elephants (a critical conservation issue) and to locally important conservation questions in the study area. I am not an expert on the specific methods used in the study, but I found them to be totally convincing, and the statistical analyses are sound. I congratulate the authors on a nice piece of work. I have minor comments, questions, and suggestions for improvement.

Introduction is succinct, well written, and well motivated.

Abstract, line 37: Is behavioral response to research vehicles a validated proxy for past/current experience? Results support the relationship between response and stress hormones, but how reliably can that be interpreted as a function of past experience as opposed to e.g. current environmental and physiological conditions? I guess you mean both of these things, but that wasn’t entirely clear to me the first time I read it.

93-96. but over a multidecadal span, elephant numbers have increased dramatically in Laikipia, haven’t they?

140-141. The red soils at Mpala and elsewhere in this part of Laikipia are not vertisols; they are classified as alfisols and differ quite dramatically from vertisols in most key properties (e.g., they are more sandy and well-drained, as opposed to clay-rich and poorly drained). Pringle et al. (2016, Large herbivores promote habitat specialization and beta diversity of African savanna trees. Ecology 97:2640-2657) provided a description of the vegetation differences between these soil types—and, importantly in the context of the present study, the importance of elephants in maintaining these vegetation differences.

Methods seem thorough and robust, although I am not an expert on the measurement and analysis of fecal hormones. Statistical approach is sound.

The Results are interesting, important, and in some cases surprising. I like the organization of this section into descriptive statistics and hypothesis testing. The model-selection results are convincing. It is unsurprising that animals that reacted fearfully to vehicles had higher FGM concentrations, although that is useful to know and a helpful sanity check on the results. It likewise stands to reason, although does not seem guaranteed, that FGM concentrations were lower in more productive (i.e., higher NDVI) conditions; in other words, that result makes ecological sense, but is still quite interesting to me.

However, the third major result—that elephants resident at Mpala had higher FGM concentrations than non-residents—is both fascinating and surprising (to me, at least). Although the spatial refuge hypothesis predicts higher stress in animals with more restricted ranges, Mpala is superficially quite a benign habitat, with generally limited poaching and harassment of elephants. The authors attribute this effect to higher levels of harassment in the north of the property, but I didn’t see the data to support this inference (perhaps I missed it?) From Fig. 1, it seems that the authors could explicitly test whether the FGM concentrations increased from north to south within Mpala, as there are plenty of samples across the north-south gradient on the property. I would recommend attempting this analysis (one would have to account for the difference in NDVI between north and south, which is a potentially confounding factor). It is certainly plausible that FGM concentrations are higher in the north, where there does tend to be an increased frequency (albeit intermittent) of human-wildlife conflict. But it would be nice to see this tested directly. The implication that British military training exercises might be stressing the elephants is intriguing and potentially important for policy. However, this seems rather speculative, as the military presence on the property is also highly intermittent, and is financially important for the property. I have no personal opinion on or insight into whether this presence is genuinely harmful to wildlife, but cessation of this activity would have significant local implications. I think that it’s important for the authors to emphasize that they are proposing this as an hypothesis that would require further work to verify. Finally, I think that the authors are remiss not to address the possibility that activity by researchers is contributing to the higher stress of resident elephants, as Mpala has much higher vehicle traffic and human presence (not to mention an arterial public road running through it) than do several neighboring properties (e.g., Ol Jogi). If researchers were a major source of stress to elephants, then one would expect this effect to be concentrated in the south, where the bulk of research traffic occurs – another reason to explicitly analyze the FGM as a function of the north-south gradient.

296. re “Elephants are known to switch diets…” See Kartzinel et al. 2019 (Covariation of diet and gut microbiome in African megafauna. PNAS 116: 23588–23593), which provides data on this point for Mpala specifically.

Figure 3. I think it’s important to show the data points themselves on this graph, unless there is some reason why that is not possible? It’s odd (and potentially misleading) to show the fitted model without the data.

·

Basic reporting

The writing is good and clear. Some minor comments on phrasing, references, and topics that should be included in the introduction under 'General comments'.
Figures should be adjusted. Results should be discussed more with the original hypotheses/arguments in mind.

Experimental design

Research questions are well defined. The research adds to a limited body of knowledge on how migratory behaviour affects FGM levels, and how both ecological and human disturbances affect FGM levels.
Methods should be explained more and some proxies used should be interpreted differently in relation to FGM levels.
The availability of the raw data and RScript are commendable.
More comments under 'General comments'.

Validity of the findings

The statistics should be redone, though this should not change the main results/findings.
some conclusion are overstated and misrepresented.
More comments under 'General comments'.

Additional comments

Review of Differing physiological and behavioral responses to 1 anthropogenic factors between resident and non-resident African elephants at Mpala Ranch, Laikipia
County, Kenya for PeerJ (#48932)

This manuscript present an interesting case-study on how anthropogenic and environmental disturbances and social behaviour can affect the physiological state of a species of high conservation interest, the African elephant. The researchers aim to investigate how 1) migratory behaviour, 2) human disturbances (estimated as behavioural reaction to car), and 3) vegetation changes affect fecal glucocorticoid metabolite (FGM) levels of elephants. They found that resident populations and those elephants with fearful reactions to cars have higher FGM levels. However, vegetation changes was the main driver of FGM, with higher FGM levels when NDVI was low. The paper presents a unique combination of measurements (behaviour and environment) that are highly relevant to current conservation physiology research. It is, for the most part, well written and clear to follow. The authors supply both the raw data and the RScript, which is uncommon yet extremely commendable. This, together with the very interesting topic of the manuscript, resulted in a rather long and thorough review on my part.
This manuscript has the potential to offer an interesting insight in factors affecting FGM levels in elephants. However, more information is needed on what defines a migrant or resident elephant. Furthermore, the behavioural reaction to car is misleading and should be discussed and/or analysed differently. More information on the NDVI data collection is desirable. Further, I have comments on the statistical approach.

With these major revisions, I think this manuscript could be a valuable contribution to the journal.



MAIN COMMENTS

1) Migrant behaviour

The authors frame the manuscript as contributing significantly to our understanding of how migratory behaviour affects the stress response in elephants. This is indeed a particularly interesting topic and little information exists on the effects of migration on GC physiology in African mammals. However, the difference between migratory and resident elephants is not clear. Namely, the authors refer to Goldenberg et al. 2016 to show some elephant families were considered resident, while others migratory, but this paper does not mention this. Furthermore, all samples (of both resident and migratory) were taken inside of the Mpala reserve and are spread out equally throughout the study area. Further still, little information about the ecology of these migratory elephants is given: e.g. is there seasonality in their movement, what is there home range (compared to residents), do they move in and out throughout the year, or stay 5 months outside of the reserve? To a naive reader, this migratory vs resident distinction is not clear and requires more information.



2) Methods

The authors use ‘reaction to the car’ as indication for human disturbance. This has been done numerous times before, but might not be a great proxy for general human disturbance. Indeed, even though the authors refer to Goldenberg et al. 2017 to back up their method, the main finding of this paper is that this proxy does not pick up on even strong stressors such as poaching pressure. Goldenberg et al. 2017 (similar study area and same study species) advise against using this kind of proxy to link to a specific disturbance! They find, however, that this reaction behaviour is different between migratory and resident populations. This could be explored by the authors in this manuscript (a Chi-sq. test shows that residents are more often calm and migrants more often terrified from approaching vehicles).

The use of NDVI as a proxy for vegetation quality presents an interesting insight in ecological drives of change, and its use is relatively new in conservation biology. Stabach et al 2015 and Hunninck et al 2020 (cited in this manuscript) both find the same effect as the present manuscript. However, their description of how they collected and analysed the data is much more detailed. To be able to correctly interpret this proxy (which has many potential issues; Pettorelli et al 2005), more info (or clearer info) on how the data is collected is needed. E.g. is the NDVI estimate the closest in time and space as was available from the online data? How does this NDVI measure relate to ecological valuable info on the foraging opportunities for elephants, which are mainly browsers, if I’m not mistaken.
Furthermore, when I regressed NDVI over Date in the author’s dataset, there was no pattern, while I expected to see a negative correlation: the vegetation should dry over the course of the dry season (or if the wet season started within the span of the study, NDVI should go up). This was found in both Hunninck et al 2020 and Stabach et al 2015. Adding a plot with mean NDVI values for the study site on the y-axis, and date (year 2019) on x-axis would be interesting and give a basis to interpret what the NDVI values mean in this system; as would a map of the mean or max annual NDVI values in the system (this would be for the appendix).

3) FGM vs timing

The link between migratory status (migrant/resident) and reaction behaviour (calm/skittish/terrified), and the FGM results is not straightforward. FGM measurement give an indication of the physiological status of an elephant since a few days before the sample was taken. Therefore, if a migrant elephant was outside of the park for 5 months but came back into the park one month before its sample was taken, then the FGM measurement does not link to the elephant having been outside of the park. This understanding is not reflected in the manuscript and should be adjusted. I also advise including a paragraph on the GC physiology and how this relates to FGM levels. This general introduction is the norm in papers using FGM levels.

4) Statistics
a. Linear regression
The authors use three different tests where only one is needed. Using only linear regression would not only improve statistical power, but also make the statistics/results more clear. This does not affect results. The command emmeans can then be used to discern statistical differences among groups.

b. Random effect
The authors use Family_ID as a random effect, but this is not needed, as only one group in that random factor has 2 observations. The authors should instead remove one of the two observations in that group and use simple linear regression models instead.

c. Confounders
The authors present several confounders but these have been tested separately and thus are statistical anomalies. Indeed, to be able to test whether e.g. Sex explains any variation in FGM, this independent variable need to be added to the initial model (i.e. FGM ~ ranging + reaction + NDVI) and then compared to the initial model without that variable (model selection). When done so, none of the confounders are significant. Only the models with time-of-day and time-delay added as a predictor are marginally better than the initial model.

5) Human disturbance proxy

The authors investigate the potential effect of human disturbance on elephant FGM levels by assessing both their migratory behaviour and behavioural reaction to a car.

Additionally, the discussion section on human disturbance suffers from circular reasoning. First the authors hypothesize that migrant elephants will have higher FGM because of higher disturbance outside Mpala. When this is found to be not the case, the authors instead state that the disturbance is in fact higher inside and therefore residents have higher FGM levels. The authors make several more unsubstantiated claims (see further).
Instead, the authors could consider using a ‘distance to nearest military base’ as a proxy for human disturbance, or, if there is information on settlements, ‘distance to nearest settlement’. This would at least allow the authors to make any statements about the potential influence of human disturbance.



MINOR COMMENTS

Line 27: “determine the impacts on reproductive output and health”. In this manuscript however, neither reproductive output or health are assessed. Rephrase so that this background section is relevant to the manuscript.

Line 31: no hypotheses are mentioned here. I would strongly advice to shorten the abstract to a readable size of 300 words, yet add the main hypotheses.

Line 37: “proxy for elephant’s past/current experience with anthropogenic disturbance”. The reaction behaviour to a vehicle is not a good proxy for past experience with general human disturbance (see Goldenberg et al. 2017). In the best case, it is a proxy for past experience with vehicles. However, it could just as well be a better proxy for personalities, regardless of past experience. This has implications for the discussion as well.

Line 45: “particularly in the north”. This implies a spatial variation in disturbance yet this is not measured.

Line 47: “affect the stable, resident inhabitants of that region”. Why would migrant elephants that are in this area for several months concurrently with resident not be affected by the human disturbances? Also, as is mentioned later, residents do move around within Mpala; why would they not move away from this disturbance?

Line 54: this introduction lacks an important paragraph on glucocorticoid physiology and application/function of FGM levels.

Line 57: “africana) (Wittemyer” should become “africana; Wittemyer”

Lines 81-101: this section is mainly about Laikipai region, and is not so relevant to the manuscript. Consider shortening or removing and adding it to study area section in methods.

Line 81: For naïve readers, when looking at the map, Laikipai is in the centre of Kenya?
Lines 82-83: “The abundance of wildlife and domestic ungulates is inversely proportional in Laikipia across three main land use types”. This sentence is not clear. I guess you mean that the more livestock there is, the less wildlife, but rephrase so this easier to understand.

Lines 91-93: “The Proportion of Illegally Killed Elephants (PIKE), a measure of the severity of illegal killing in monitoring sites, has increased over the past decade, peaking at 70% of all deaths in 2012”. This data is outdated and should either be updated or thoroughly rephrased, as the PIKE has been going down since 2012 in most areas.

Line 102: remove ‘therefore’

Line 105: “physiological demands of ranging over long distances”. I agree with this statement but the authors abandon the reasoning/arguments of this entire section in the discussion because the results do not fit these arguments. The discussion needs to be adjusted to reflect these arguments accordingly.

Line 113: “spatial refuge hypothesis”. Here, a specific hypothesis is proposed as a fundament to base the first prediction on (i.e. migrant elephants have higher FGM), yet this is the only time this hypothesis is mentioned. The discussion needs to be changed so that a clear argumentation is given as to why this hypothesis does not hold.

Line 125: refrain from using the term ‘stress levels’ but instead use ‘FGM levels’

Line 125: “long distance non-resident”. Please do not use different synonyms throughout the paper. You clearly defined what resident elephant and migrant elephant mean in this context, and these terms should be used throughout the manuscript.

Line 127: neither ‘vegetation’ nor ‘season’ are mentioned in the introduction, yet an entire prediction is based on it. This needs its own paragraph in the introduction.

Line 132: please include a section on elephant ecology. Specifically their movement behaviour, foraging behaviour
Line 133: please include information on the seasons in this area. When are the wet/dry seasons, how long do they last and what is the average rainfall?

Line 150: “classified as residents or non-residents based on the temporal use patterns of the ranch”. This needs more information. What is their temporal use? How long do migrants leave the Mpala area? Do they come and go, or stay away for longer times on end?

Line 153: “home ranges are restricted to one conservancy”. Such as Mpala? Make this clearer and specific to the study animals.

Line 157: “Goldenberg et al 2016”. This paper does not mention this nor gives information on this topic.

Line 171: “Goldenberg et al 2017”. This paper states specifically that ‘reaction behaviour to vehicle’ is not a good proxy for human disturbance, not even for much stronger disturbances such as poaching. However, the paper does state that this behaviour is different between residents and migrants; this is also the case for the elephants in the current manuscript. This could be added to the paper as it is an interesting finding, in my opinion, and could relate to a behavioural syndrome or personality.

Line 180: this section needs more information on how the data was exactly collected and also on how this NDVI measure relates to the ecology of the African elephant. Elephants are predominantly browsers and would be less affected by grasses drying over the dry season, which is apparently what is mainly measured by NDVI (see Hunninck et al 202, Stabach et al 2015).

Line 191: “within 1 hour of defecation”. But later the authors state that samples were collected within “2-30 min” (line 196). From the dataset, I see that the latter is correct.

Line 199: “Fecal samples were extracted similar to that described previously”. This should be rephrased to include that the method ‘was based on Wasser et al.’ as the current extraction method is quite different from Wasser et al.

Line 210: the EIA analytical validation seems to be performed well, but data and statistics are lacking. This data and the statistics should be included in the dataset and R script, respectively.

Lines 213-215: this is a very strange biological validation. Not in the least because it is difficult to do decent statistics on this. For every mean ± SD that is given, where is the N? for every statement of significance, where are the test statistics? The example of tail injury and lameness seem to come from different elephants and no baseline is given; single values cannot be compared.

Lines 215-217: did you analyse the same samples several times with different assays? If so, how many samples were analysed several times (i.e. what is this correlation based on?). also, why not analyse all samples with the validated assay?

Line 221: it would be much more beneficial to present the data as mean ± SD not standard error. Also please put the first and second sentence of the paragraph at the end of the paragraph.

Line 222: RStudio is not referenced correctly here.

Line 222: remove “i.e.” and “first”

Line 223: no linear mixed models should be used here as the random effect only has one group with more than one observation. Simple linear regression is applicable here. I advise removing one of those observations within that group (instead of a sample size of 156, it would now become 155).

Lines 225-227: “A stepwise criteria using p value from the package lmerTest (Kuznetsova et al., 2017) was used to compare different models (Table 2).” No model selection should be considered here. The hypotheses are clear and call for an additive model structure with the three main predictors. Model selection should be done when considering the confounders.

Line 228: when using ∆AICc, you should add the threshold criterium for deciding which model is better/equal. This is usually set at ∆AICc < 2.

Line 230: “that explained most of the variation in the data”. Rephrase to ‘that best fit the data’.

Line 231: validation of the simple linear mixed model can be done with the R base ‘plot’ command. These 4 plots could be added in the appendix.

Line 236: as mentioned, to prevent the issues with multiple testing and inflating the Type-I error, the effect of confounding factors should be assessed by adding them to the basic/initial model i.e. log(FGM) ~ ranging + reaction + NDVI. Then model selection should be done on the basic model and the basic model + confounder to see if the addition of the confounder better explains the variation in the data.

Line 238: the results need to be updated according to the updated statistics.

Line 250: I do not understand why the authors have split the result section up and why there is a separate hypothesis section. All results should be mention in a logical fashion (main results first, minor results after) within the results section.

Line 257: suddenly there is a result about an interaction model, while this is not mentioned in the predictions or in the methods/statistics section. This interaction model should wither be removed or properly introduced and argued for.

Line 285: “exposure of resident elephants to higher levels of human activities in the northern part of Mpala”. Yet both resident and migrant elephants were sampled all over Mpala. This spatial effect in human disturbance can be assessed by e.g. looking at potential effect of ‘distance-to-nearest-military_base/settlement’ on FGM levels.

Line 288-290: this potential disturbance should also affect migrants, no?

Line 293: more examples should be added here.

Line 300: “non-residents having access to higher quality diets due to greater foraging”. What is meant by ‘greater foraging’ and why would migrants have this over residents? Also, you have NDVI data that is supposedly measuring vegetation quality. Could you then not check whether vegetation quality is higher inside vs outside Mpala?

Lines 303-305: “Together, our results agree numerous studies showing significant effects of human disturbances on FGM in elephants”. Your results do not show this. You merely assume that these high levels of FGM in residents is due to human disturbance but have no proof of that. You also propose other mechanisms, such as vegetation quality. This sentence needs to be rephrased.

Lines 311-312: “an indication that past/presence negative interaction with humans could be influencing the behavioral and physiological state of elephants”. Behavioural reaction to a vehicle does not necessarily offer insight in past experiences of elephants, and especially does not mean that elephants have had negative experiences with ‘humans’ in general. The differences in behavioural reaction could be due to personality differences and be unaffected by past experiences with humans.

Line 313: “This was particularly true for resident elephant”. No interaction with ‘ranging behaviour’ was done during the modelling, so the phrase ‘particularly with’ cannot be used here.

Line 313: “that were exposed greater rates of disturbance”. This was first assumed not to be the case, then proposed as a potential explanation, and here it is presented as fact? Please rephrase thoroughly. Include “to” in between ‘exposed’ and ‘greater’.

Line 314: “The landscape of fear paradigm has recently emerged”. This concept have been around for many years. Please rephrase.

Line 318: “Our study corroborates others in ungulates”. No evidence of an effect of human disturbance was found in this study, only assumed, therefore it does not corroborate previous findings. Please rephrase.

Line 327: “only slowly to within 20 m of the closest elephant”. I feel that the non-invasiveness of this method is underestimated here. First, even when approaching slowly, weary elephants will have noticed the vehicle from hundreds of meters away, and second, approaching up to 20m is very close to an elephant.

Lines 327-329: “Even still, many Mpala elephants had reactions to the approach of these vehicles, both in terms of behavior and FGM.” This is incorrect: elephants did not exhibit elevated FGM levels in response to approaching vehicles. FGM levels were higher in animals that reacted fearful to approaching vehicles; as discussed before, this can mean many things. This is very different from an GC response to vehicles. Even so, approaching vehicles would present an acute stressor, something that would not be visible in FGM (this needs to be in the introduction).

Line 343: the reference Pokharel et al 2019, is not relevant here. Better use previously cited Stabach et al 2015 or Hunninck et al 2020.

Line 349: “(end of April to mid-May)”. This should be mentioned in the methods. If the dry season only lasts until mid-May, then NDVI values should increase dramatically after that, yet this is not seen in the data.

Lines 355-396: this section should be updated according to the new analyses. Most of the topics can still be there, but your results show no effect of any of the selected confounders.


FIGURES & TABLES

Figure 1
- KLEE, SIGEO, Uhuru plots are shown but not used in the paper? This should then be removed.
- Although the elevation base layer is interesting, since it is not mentioned in the paper and only differs in max 250 m, I would suggest you used a annual mean (or max) NDVI base layer instead.
Figure 2
- This figure should be removed as it does not relate to the model results and adds no information.
Figure 3
- Please show the raw values (datapoints) as well.
- The line that is plotted is unlikely to be the regression line. A regression line would be one straight line (if on the original scale) or a smooth line with decreasingly steep slope, while this line is in two parts and is likely just a line fitted to the datapoints

SUPPLEMENTAL INFORMATION

Figures S3, S4, S5, S6, S8 are not relevant anymore and should not be included.
Figure S7 is presumably FGM ~ Group size. This is similarly not relevant and should be removed

---

## Round 0.2 · accepted · Accept

Thank you for a very nice submission and an excellent revision. All the previous reviewers are satisfied with your changes to the MS and I agree with them that this will make a nice addition to a sparse literature. Please attend to the very minor edits mentioned by Reviewer 1 during proofing.

Reviewer 1 ·

Basic reporting

Professional English: Acceptable. The paper is quite well-written.

References & background: Appropriate and up-to-date.

Professional article structure w raw data: Clear and classic structure; raw data appear well organized and understandable

Self-contained unit of publication: Yes

Experimental design

Within aims & scope of journal: Yes

Research question well defined, relevant & meaningful, fills a gap: Yes

Rigorous, high technical standard, high ethical standard: Yes - this is quite a strong study.

Methods described in sufficient detail: Yes, the revision has added sufficient detail

Validity of the findings

Data provided; robust, statistically sound, controlled: Yes

Conclusions well stated, linked to original question, limited to results: Yes, the revision has sufficiently addressed the issues noted in the original submission

Speculation is identified as such: Yes

Additional comments

This is a thorough revision that sufficiently addresses all of my prior concerns. Thank you for a well-executed and very interesting study that will certainly be a welcome addition to the field of wildlife stress assessment generally, and elephant management specifically.

I noticed a few tiny, typo-level issues that you could address during the copy-editing phase, listed below.

line 91 - "GCs" is used here but this abbreviation has not been defined yet in the main text (it is defined in the Abstract, but the main text needs to stand on its own). Similarly, define "FGM" when first used in the main text

line 239 - "samples" should be "sample." Also, suggest adding "each of" - "One fecal sample was collected from each of 156 uniquely identified elephants..."

line 277 - add "a" ("A linear...")

line 279 - change "factor" to "factors"

line 286 - remove extra parenthesis before Burnham

line 301 - equals sign is missing for N=38

line 335 - typo, change NVDI to NDVI

lines 381 & 423 - move the parenthesis that is in front of author name

line 452 - delete "a" before "calves"

line 498 - delete "the" at end of sentence

line 505 - delete "a" (change to "need for more comprehensive")

Reviewer 2 ·

Basic reporting

Good.

Experimental design

Good.

Validity of the findings

Good.

Additional comments

The authors have satisfactorily addressed my questions and comments on the previous version and I am also impressed at the thoroughness and care with which they have addressed the comments from the other reviewers.

·

Basic reporting

The manuscript is clear, easy-to-follow, well-structured, and very well written. Background information on all relevant topics is given, results are discussed in detail and with consideration for the studies shortcomings and other potential hypotheses. The Figures and tables are excellent: to the point, informative, clear to understand. Hypotheses are well defined and perfectly addressed in the discussion. Overall, and excellently written manuscript.

Experimental design

The statistical analyses have now been updated and are on-point. Concise, clear, well-explained, rigorous, and parsimonious. The addition of the raw data and RScript offers a standard of transparency that is exemplary.

Validity of the findings

The findings have now been discussed in a much more detailed and fair way, including multiple potential mechanisms explaining the results, and acknowledging the study's shortcomings.

Additional comments

Dear authors,

Thank you very much for thoroughly revising your manuscript and excellently addressing my comments. The changes made answer all of my comments more than I could have expected. The manuscript is well-written and a very interesting contribution to our understanding of GC physiology. I have no further comments. Thank you for your excellent contribution.

Kind regards,
Louis Hunninck